# N-3 Polyunsaturated Fatty Acids and Their Lipid Mediators as A Potential Immune–Nutritional Intervention: A Molecular and Clinical View in Hepatic Disease and Other Non-Communicable Illnesses

**DOI:** 10.3390/nu13103384

**Published:** 2021-09-26

**Authors:** Francisca Herrera Vielma, Rodrigo Valenzuela, Luis A. Videla, Jessica Zúñiga-Hernández

**Affiliations:** 1Department of Biomedical Basic Sciences, School of Health Sciences, University of Talca, Talca 3460000, Chile; francisca.herrera@utalca.cl; 2Department of Nutrition, Faculty of Medicine, University of Chile, Santiago 8380000, Chile; rvalenzuelab@med.uchile.cl; 3Molecular and Clinical Pharmacology Program, Institute of Biomedical Science, Faculty of Medicine, University of Chile, Santiago 8380000, Chile; lvidela1944@gmail.com

**Keywords:** inflammation, N-3 PUFAs, immunonutrition, liver disease, clinical trials

## Abstract

In recent years, the beneficial effect of n-3 polyunsaturated fatty acids (n-3 PUFAs) intake on human health has been widely accepted in the field of immunonutrition. Today, we find a diversity of supplements based on n-3 PUFAs and/or minerals, vitamins and other substances. The main objective of this review is to discuss the importance of n-3 PUFAs and their derivatives on immunity and inflammatory status related to liver disease and other non-communicable illnesses. Based on the burden of liver diseases in 2019, more than two million people die from liver pathologies per year worldwide, because it is the organ most exposed to agents such as viruses, toxins and medications. Consequently, research conducted on n-3 PUFAs for liver disease has been gaining prominence with encouraging results, given that these fatty acids have anti-inflammatory and cytoprotective effects. In addition, it has been described that n-3 PUFAs are converted into a novel species of lipid intermediaries, specialized pro-resolving mediators (SPMs). At specific levels, SPMs improve the termination of inflammation as well as the repairing and regeneration of tissues, but they are deregulated in liver disease. Since evidence is still insufficient to carry out pharmacological trials to benefit the resolution of acute inflammation in non-communicable diseases, there remains a call for continuing preclinical and clinical research to better understand SPM actions and outcomes.

## 1. Introduction

Inflammation is a response of the immune system that allows survival during an infection or injury, eliminating harmful stimuli and initiating the recovery process; therefore, inflammation is a vital defense mechanism for health [1]. During acute inflammatory responses, cellular and molecular events and interactions minimize impending injury or infection, processes that contribute to tissue homeostasis and resolution of acute inflammation [2]. However, uncontrolled acute inflammation may become chronic and lead to a variety of chronic inflammatory diseases including cardiovascular diseases, autoimmune diseases, fibrosis and cancer [1]. Currently, the resolution of inflammation is recognized as a dynamic, programmed and self-limited process, where its regulation is obtained through mechanisms of action based on endogenous anti-inflammatory and/or pro-resolutive agonists, called specific pro-resolving mediators (SPMs) [2].

Immunonutrition is the subject that deals with the study of the interactions between nutrition and immunity in its entirety, that is, the immune system, infections, inflammation and injury or tissue damage [3]. An adequate nutritional state is synonymous with a good defense function of an organism. Currently, scientific evidence has confirmed the crucial importance of dietary intake and its role in regulating the individual’s defenses, as well as in the risk of developing acute and chronic diseases [4], such as chronic liver disease [4,5]. In the context of liver disease, whose etiology can be generated by different agents such as viruses, toxins and drugs, is becoming one of the most serious public health problems [5]. Liver inflammation is a common trigger for liver disease and is considered the main driver of damage to liver tissue, causing the progression of non-alcoholic fatty liver disease (NAFLD) to severe fibrogenesis and, ultimately, hepatocellular carcinoma (HCC) [6].

Immunonutrition usually involves arginine, nucleotides and n-3 polyunsaturated fatty acids (n-3 PUFAs) [7]. Previously, the beneficial effects of n-3 PUFAs on the stress response and immune function it has been demonstrated in preclinical and clinical studies [8]. Additionally, n-3 PUFA supplementation has been used as a nutritional complementary approach in liver transplantation [9] and hepatectomy studies [10], with promising results related to a reduction in postoperative complications and an improvement in postoperative nutritional state. The latter aspect is of particular importance due to the protein deficiency and general malnutrition in patients with advanced chronic liver disease (CLD), which in the case of cirrhosis reaches rates of 50% to 90% of patients [11]. In the present work we focus on describing the role of n-3 PUFAs and their derivatives as immunonutrition in liver diseases; we specifically delve into the potential of SPMs as immunomodulatory and nutritional supplements related to complications in liver surgery, resections and CLD.

## 2. Methods

The review includes studies that considered the molecular pathways and beneficial effects of n-3 PUFAs, particularly EPA and DHA, and lipid mediators from EPA and DHA using in vivo and in vitro models of nutritional intervention in liver disease. Study searches were performed using the PubMed database from the National Library of Medicine—National Institutes of Health, the World Health Organization (WHO) and clinical trials from www.clinicaltrials.gov-U.S (accessed from March to September 2021), among others.

## 3. Nutritional States in Liver Diseases

According to the 2019 burden of the liver disease in the world, more than two million persons die of liver disease per year worldwide, half of them related to cirrhosis complications and the other half due to viral hepatitis and HCC [12]. Spontaneous survival related to ALF rates depends on the underlying disease, and there are reported ranges from 17% to 68%; however, when liver transplantation became available, survival rates have been reported at ranges from 60% to 79% [13,14,15]. Among the liver disease types, alcohol-associated liver disease (AALD) stands out as the major global cause of liver disease, since over 50% of mortality related to cirrhosis is attributable to alcohol [12]. Progression to cirrhosis is higher in patients with AALD compared to NAFLD. NAFLD is also related to two differential conditions: non-alcoholic fatty liver (NAFL) and non-alcoholic steatohepatitis (NASH), a condition that includes degrees of fibrosis, cirrhosis and HCC [12,13]. It should be considered that obesity could potentiate the increase in both AALD and NAFLD rates in the following years. Among the other prevalent liver diseases, it is important to mention (i) viral hepatitis, which is associated with 1.34 million deaths and affects principally low- and middle-income countries; (ii) primary sclerosing cholangitis and the related cholangiocarcinoma, gallbladder and colorectal cancer; (iii) primary biliary cholangitis; and (iv) autoimmune hepatitis, among other less prevalent pathologies [12,13,15].

One of the clinical challenges in liver disease is to normalize, revert or at least ameliorate the malnutrition status related to the chronicity and liver failure. Where malnutrition (undernutrition) is frequently a burden in patients with liver cirrhosis, occurring in 20–50% of them, it is associated with the progression of liver failure. Malnutrition has been reported in 20% of patients with compensated cirrhosis and in more than 50% of patients with decompensated liver disease [16]. In addition to undernutrition, overweight and/or obesity are observed in cirrhotic NASH patients. Muscle mass depletion (sarcopenia) may also occur in these patients, but it coexists with obesity and it might be overlooked. Obesity and sarcopenic obesity may worsen the prognosis of patients with liver cirrhosis [16,17]. 

In terms of AALD, nearly all patients with cirrhosis are malnourished and the degree of malnutrition correlates with disease severity and their complications. The malnutrition is multifactorial and is related in part to poor dietary intake (anorexia and dysgeusia), magnesium deficiency, disproportionate caloric intake from alcohol and the subsequent micronutrient and vitamin deficit in addition to impairment of gastrointestinal mucosal absorption caused by alcohol, among other factors [18]. Most of the micronutrients described as part of malnutrition are folate, vitamin B6, vitamin A, thiamine and minerals such as selenium, zinc, copper and magnesium, which in some instances are thought to be involved in its pathogenesis [19]. According to the current guidelines of the American Association for the Study of Liver Diseases (AASLD), all patients with alcoholic hepatitis or advanced ALD should be assessed for nutritional deficiencies and treated aggressively with enteral nutritional therapy [19,20]. 

## 4. Immunonutrition

Nutrients are initially recommended to provide energy, protein and micronutrients that are essential to prevent muscle fatigue and to avoid hunger-induced immune exhaustion. Currently, the field of nutritional support therapy has evolved since its conception in the eighties to immunonutrition, defined as the use of specific nutrients to increase the immune response and support trauma in the period of critical illness and stress, administered enterally or parenterally [21]. Immunonutrition has been incorporated largely into studies to improve the clinical course of critically ill and surgical patients, where they frequently require an exogenous supply of nutrients [22].

Historically, immunonutrition systems of administration are: (i) Enteral nutrition (EN), a nutritional support technique, in which nutritional substances are supplied to the gastrointestinal tract, distally to the oral cavity, nasally via implanted tubes or an enterostomy [23]. There is evidence that the enteral route has obtained more benefits and improvements in patients over the administration of parenteral nutrition and it ought to be preferred whenever there is a functioning gastrointestinal tract [24], EN being a safer and less expensive technique [25]. (ii) Parenteral nutrition (PN) provides lifesaving nutritional support in situations where the parenteral supply of calories does not meet the body’s demands or when EN is not possible to use for some reasons. PN preserves lean body mass, supports immune functions and reduces metabolic complications in patients who would otherwise be unable to feed. However, PN associated with intestinal failure exhibits adverse effects, such as deterioration of hepatic function [26], which normalized after the interruption of the procedure; therefore, it can represent a serious risk factor in people who receive this treatment in the long term [27]. The basis of a correct immunonutrition is to obtain the maximum benefits of this technique that decrease the days spent in hospital, diminish the incidence of infections and reduce hospital costs. The effects are more consistent in patients with severe trauma, including burned subjects, some critical pathologies, those undergoing a major surgical procedure, polytraumatized patients or those that in their processes generate a high level of organic stress, increasing nutritional requirements, especially malnourished individuals prior to a surgical event [28]. Nutrients that have been identified to generate an immune benefit include: (i) Glutamine, the most abundant non-essential amino acid, which is essential for fast-growing enterocytes and cells of the immune system [29], restores the immune depletion of critically ill patients [30] and regenerates the liver after weight gain and after hepatectomy [31]. (ii) Arginine, a non-essential amino acid that becomes essential during states of stress and critical illness [32]. This amino acid has been shown to be of vital importance in vasodilation, calcium and hormone release, neurotransmission, cell proliferation, secretory activities and immunity [33]. In the context of the liver, arginine has a regenerative effect after a hepatectomy [34], hepatoprotective properties against lipid peroxidation [35] and improves injury by preservation of liver transplantation [9]. (iii) Nucleotides, the endogenous, biologically active substances that play an important role in almost all biochemical processes as they are the precursors of DNA, RNA, ATP and coenzymes, among others. Dietary nucleotides may be necessary for the optimal growth and function of active metabolic cells, such as lymphocytes, macrophages and intestinal cells [36]. In this context, they contribute to correct the alterations in the levels of fatty acids in plasma and liver microsomes from rats with hepatic cirrhosis [37,38], besides acting as regenerative compounds in injured cirrhotic livers [39]. (iv) Finally, n-3 PUFAs are important for the structure and function of cell membranes and have additional important functions, such as being precursors of SPMs [40]. They are involved in physiological processes that modulate the state of health and the onset of chronic diseases, such as the regulation of plasma lipid levels, cardiovascular [41] and immune functions [42], neuronal development [43], visual activity, diabetes [44], rheumatoid arthritis [45,46] and cancer [47], in addition to the positive effects on the immune status of critically ill patients [48]. 

## 5. N-3 PUFAs and Inflammation 

For some metabolic and structural functions, n-3 PUFAs and n-6 PUFAs undergo processes of elongation and desaturation, mainly in the liver [49]. In this way, linoleic acid (C18:2n-6; LA) from the n-6 PUFA series gives rise to arachidonic acid (C20:4n-6; AA), whereas α-linolenic acid (C18:3n-3; ALA) from the n-3 PUFA series generates eicosapentaenoic acid (C20:5n-3; EPA) and docosahexaenoic acid (C22:6n-3; DHA). AA, DHA and EPA are structural components of cell membranes, where they are released by the activation of the enzyme phospholipase A2 (PLA2) during the early stages of an inflammatory process. AA is metabolized by the enzymes lipoxygenase (LOX) and cyclooxygenase (COX), generating bioactive eicosanoids, such as prostaglandins (PG), leukotrienes (LT) and thromboxanes (TX) [50]. These include PGE2, TXA2 and LTB4, which are potent mediators of inflammation, fever, pain, increased vascular permeability [51], the release of inflammatory cytokines and enhanced activity of immune cells [52]. Instead, EPA and DHA give rise to anti-inflammatory and SPM derivatives such as resolvins, protectins and maresins [53] (Figure 1). A human diet rich in EPA and DHA allows for an increase in these fatty acids in cell membranes and reduces the AA content due to competition for the desaturation enzymes, decreasing the generation of pro-inflammatory products derived from n-6 fatty acids [51]. The molecular mechanisms involved in the anti-inflammatory effect of n-3 PUFAs in the liver are related to the downregulation of pro-inflammatory transcription factor nuclear factor-κB (NF-κB). These include (i) the formation of SPMs; (ii) binding to peroxisome-proliferator-activated receptor-α (PPAR-α) to form an inactive complex with NF-κB-p60; (iii) binding to G-protein coupled receptor 120 (GPR120) to abrogate NF-κB activation and derived inflammasome NLRP3 expression; and (iv) the formation of J3-isoprostanes, oxidation products that activate nuclear factor erythroid-2-related factor 2 (Nrf2), which induces an antioxidant response blocking the redox activation of NF-κB [54]. SPMs regulate the production of inflammatory mediators (chemokines, cytokines and inflammatory eicosanoids), preventing leukocyte infiltration and triggering macrophage phenotype changes. Thus, SPMs prevent inflammation from getting out of control and enable its termination. Furthermore, they have been shown to promote tissue repair and regeneration [55]. Furthermore, SPMs play a protective role in chronic inflammatory diseases such as atherosclerosis [56], cardiovascular disease [57], rheumatoid arthritis [58], cystic fibrosis [59], asthma [60], cancer [61,62] and periodontitis [63], and protect the liver cells against genotoxic damage and oxidative stress [64], among others. The discovery of SPMs provides a new perspective in the treatment of inflammatory disorders. These molecules are a group of endogenous chemical mediators biosynthesized in a transcellular way, and they were identified in inflammatory exudates. Within this group we find three distinctive families, namely resolvins, protectins and maresins [55].

### 5.1. Resolvins

These SPMs were the first described resolving molecules, and were identified in exudates obtained during spontaneous resolution, in a murine model of inflammation based on the formation of air pockets in the dorsal area of skin, which spontaneously resolve [65]. Their name derives from their ability to regulate resolution through multiple mechanisms, including the prevention of neutrophil infiltration, promotion of apoptotic neutrophil phagocytosis to clarify the lesion and enhancement of the resolution of inflammation within the lesion and thus promoting tissue regeneration [66]. In this way, the resolution interval is shortened, preventing the progression of an acute to chronic inflammatory response [67]. These lipid mediators have been shown to have anti-inflammatory, anti-fibrotic, anti-angiogenic and infection resolution properties [68].

### 5.2. Protectins 

These molecules were initially described in human respiratory tracts and called D series protectins, because they derived from DHA and were later converted by leukocytes into protectin D1 or neuroprotectin, which protects neuronal tissues [69,70]. Protectin D1 is also produced by lymphocytes in the peripheral blood, and its action consists in reducing tumor necrosis factor-α (TNF-α) and interferon gamma (INFγ) secretion as well as blocking the migration of T cells, promoting apoptosis [71].

### 5.3. Maresins 

The most recently discovered bioactive compound family are maresins, synthesized from DHA by macrophages that relate to the resolution of inflammation [41,72]. Their function is to limit tissue infiltration of polymorphonuclear (PMN) leukocytes, reduce collateral tissue damage by phagocytes, shorten the resolution interval, improve phagocytosis and counter-regulate pro-inflammatory chemical mediators. In addition, mitigation of hepatic steatosis is observed in obese mice [73] and they exert protection against ischemia–reperfusion (IR) liver injury [74]. 

## 6. Beneficial Effects of N-3 PUFAs

Currently, n-3 PUFAs are considered essential for normal growth and development, and these fatty acids also play an important role in the prevention and treatment of various diseases, as exemplified in the followed paragraphs.

### 6.1. Cardiovascular Health 

Cardiovascular diseases are among the two leading causes of death in the world [75]. The n-3 PUFAs were shown to have numerous heart health benefits, preventing atherosclerosis by reducing total cholesterol concentrations, triglycerides [76] and blood pressure in patients with hypertension [77]. Besides, the administration of n-3 PUFAs increases high-density lipoprotein (HDL) cholesterol levels [78], prevents platelet clumping [79] and they have the potential to beneficially impact arterial wall remodeling and cardiovascular outcomes by targeting arterial wall hardening and endothelial dysfunction [80].

### 6.2. Nervous System 

For several years, in vivo and in vitro studies with n-3 PUFA supplementation in humans revealed a neuroprotective effect, especially in lesions induced by ischemia and by excitotoxicity produced by neurotransmitters [25]. The administration of DHA prevented the anatomical functional deterioration in neurons of animals, characteristic of diabetic neuropathy, in addition to diminishing oxidative damage and learning problems in rats with traumatic brain injury [51]. It has also been determined that they can correct visual and brain problems in patients with demonstrated deficiency. Besides, n-3 PUFAs are part of the treatment of multiple neuropathologies, such as attention deficit hyperactivity disorder, multiple sclerosis, depression, schizophrenia and Alzheimer’s disease [81]. 

### 6.3. Cancer

Cancer is one of the leading causes of death in the Western world. The study of the use of n-3 PUFAs in cancer and cancer-related cachexia is of great interest. There are studies in mice and cell cultures showing that the administration of EPA and DHA can control the initiation and progression of prostate cancer [82]; specifically, DHA suppresses the NF-κB prosurvival pathway by arresting the growth of prostate cancer [83]. In rats with colon cancer, n-3 PUFAs directly affect the expression of genes related to tumorigenesis and apoptosis, reducing cellular and DNA damage [84]. Along with this, the study of cachexia, which generates loss of weight and tissue (mainly muscle) in cancer patients and is an indicator of a poor prognosis, revealed that the use of n-3 PUFAs supplements may increase appetite, decrease weight loss and augment lean mass, thus providing a better quality of life [82,85]. This is due to the fact that n-3 PUFAs decrease the levels of pro-inflammatory cytokines in plasma, benefiting the symptoms of patients with cancer-related cachexia that show elevated levels of inflammatory mediators [52]. In addition, n-3 PUFAs prevent breast cancer [86], skin cancer [87] and improve postoperative recovery in cirrhotic patients with liver cancer [88]. In hepatocellular carcinoma, Lim et al. (2009) demonstrated that n-3 PUFAs induce the degradation of β-catenin, a molecule related to cellular proliferation, which is associated with Wnt tumorigenesis pathways, where Wnt/β-catenin activation occurs in approximately 30% of HCC [89]. In patients with liver metastases due to bowel cancer having recurrence of the disease and death, the administration of n-3 PUFAs before liver surgery favors survival and lessens recurrence of the disease, outcomes also observed in liver metastasis from colorectal cancer [90].

### 6.4. Inflammatory Bowel Disease, Rheumatoid Arthritis and Other Autoimmune Diseases

N-3 PUFAs may contribute positively to intestinal pathologies [91]. For example, interleukins (IL)-1, IL-6 and TNF-α play an important role in the development of inflammatory bowel pathology, where the effects of n-3 PUFAs reduce pro-inflammatory cytokine levels in rat ulcerative colitis, improving several markers of cellular damage [91,92].

Rheumatoid arthritis, which afflicts 1% of the population worldwide, is a systemic autoimmune disease characterized by chronic inflammation of the synovial tissue and joint destruction. Treatment consists of anti-inflammatory and immunosuppressive drugs, which have unfavorable long-term side effects [93]. For this reason, studies have been conducted with the administration of n-3 PUFAs, reducing serum levels of IL-1, IL-2, IL-6, IL-8 and TNF-α [94]. They improve the symptoms of this disease and reduce joint tension and stiffness [45]. This supports the use of n-3 PUFAs as part of the treatment of autoimmune diseases, especially in the early stages. In addition, there have been studies in which n-3 PUFAs have benefits for other immune-related diseases, such as type II diabetes, metabolic syndrome and obesity, among others [95]. In patients with inflammatory bowel diseases, n-3 PUFAs induced a reduction in relapse rates, an improvement in symptoms and a diminution in oxidative stress in the intestine [51], and in psoriasis reduced the area of psoriasis, the severity index and the dermatological quality of life index [96].

### 6.5. Ischemia–Reperfusion-Induced Damage (IR)

There are numerous situations in which ischemia of organ/tissues can occur, with a lack or decrease in blood supply, oxygen and nutrients [97]. The IR damage is produced in situations such as myocardial infarction, acute kidney damage, systemic shock and major trauma related to surgeries, among others [97,98]. In the liver, IR damage is associated with transplantation, liver resection and trauma, where n-3 PUFA administration has had anti-inflammatory and cytoprotective effects, since it reduces the inflammatory response and increases antioxidant activity. In addition, n-3 PUFAs inhibit the translocation of NF-κB from cytosol to the nucleus [99,100], reducing the expression of pro-inflammatory cytokines through the activation of PPAR-α, which forms inactive complexes with NF-κBp65 to generate anti-inflammatory and cytoprotective effects [101].

## 7. Beneficial Effects of N-3 PUFA in Liver Diseases

The liver is especially sensitive to changes in the dietary fatty acids, and is an organ affected by diets with high levels of saturated fatty acids, *trans* fatty acids and other dietary components that can saturate its detoxifying functions, leading to metabolic disorders and liver damage. Supplementation with n-3 PUFAs is a focus of interest given their cytoprotective effects. Research conducted for liver disease has been gaining momentum recently, with the encouraging preliminary studies discussed below (Figure 2).

### 7.1. Evidence for n-3 PUFA Effects in Animal Models of Liver Disease

In non-alcoholic fatty liver disease (NAFLD), the potentially beneficial effects of n-3 PUFAs are provided by animal studies, using several models that closely resemble the pathogenesis and metabolic changes of the disease, although the dosages of n-3 PUFAs supplied are greater than those used in the human studies. The molecular mechanisms of n-3 PUFAs on liver lipid metabolism leading to an anti-steatotic response include (i) the activation of the PPAR-α/FGF21/AMPK/PGC-1α signaling cascade to increase FA oxidation; (ii) diminution in the processing of lipogenic factor sterol responsive element-binding protein 1c (SREBP-1c); (iii) downregulation of the expression of the citrate carrier reducing the cytosolic availability of acetyl-CoA units for de novo lipogenesis; and (iv) abrogation of endoplasmic reticulum stress and the associated expression of lipogenic enzymes [54,102]. Sekiya et al. (2003) demonstrated that the administration of n-3 PUFAs, in the obesity model of leptin-deficient ob/ob mice, significantly decreased the triglyceride (TG) and alanine aminotransferase (ALT) serum levels [103]. Furthermore, the authors found improvements in hyperglycemia, hyperinsulinemia and the activation of PPAR-α, promoting fatty acid (FA) oxidation, with the concomitant suppression of lipogenic SREBP-1c, concluding that n-3 PUFAs improve symptoms associated with obesity, such as liver steatosis and insulin resistance [103]. In addition, similar results were observed in n-3-PUFA-fed rats in relation to the loss of hepatic TGs and weight [104]. Besides, the administration of n-3 PUFAs by parenteral nutrition of C57BL6 mice protected the liver against hepatic steatosis with improvement in liver architecture, decreased aspartate aminotransferase (AST) and ALT serum levels, an amelioration of liver fat content and a reversion of the essential fatty acid’s deficiency [105]. Interestingly, n-3 PUFAs had insulin sensitization actions in adipose tissue and liver, with an improved insulin tolerance in ob/ob mice, in addition to an enhancement in adiponectin levels and alleviation of liver steatosis, via the formation of resolvins and protectins [106]. In relation to lipid metabolism, Shang et al. (2017) studied the effects of dietary supplementation of different DHA/EPA ratios on a high-fat-diet-induced lipid metabolism disorder and the concurrent liver damage [107]. Under these conditions, decreased body weight and lipid profiles in serum and liver were observed, concomitantly with attenuation of liver steatosis, inflammatory reactions (lower serum TNF-α, IL-1β and IL-6 levels), lipid peroxidation and the protein expression of factors related to adipogenesis and inflammation [107,108]. In addition, a lower DHA/EPA ratio appears to be more beneficial in alleviating high-fat-diet-induced liver damage in mice and lowering inflammatory risk factors [108]. In other models of liver disease, n-3 PUFAs have been shown to decrease levels of γ-glutamyl transpeptidase (γ-GT), high-density lipoprotein (HDL), albumin and total protein in liver fibrosis induced by thioacetamide [109]. Compared to a lard diet, fish oil intake significantly decreased body weight gain, white blood cell count, levels of TG, total cholesterol, fat accumulation, HDL, oxidative stress and inflammatory cytokines [108]. Furthermore, it was demonstrated that n-3 PUFA supplementation by parenteral nutrition inhibits oxidative stress in an experimental model of liver regeneration [110], reduces liver fibrosis, promotes liver regeneration, even under cirrhotic conditions [111], and suppresses liver tumorigenesis [111,112,113].

### 7.2. Clinical Trials with n-3 PUFAs in NAFLD and Its Progressive Disease States

There is a bulk of information related to n-3 PUFA supplementation and PN on liver disease [88,89,90,91,92,93,94,95,96,97,98,99,100,101,102,103,104,105,106,107,108,109,110,111,112,113,114,115,116], with a significant number of clinical trials developed or in testing state (Table 1 and Appendix A), showing that n-3 PUFA supplementation is a noteworthy field in medicine.

#### 7.2.1. NAFLD

The efficacy of prolonged n-3 PUFA supplementation was evaluated in patients with NAFLD, the first study being conducted in humans by Capanni et al. (2006), where they found that after 12 months with treatment, the Doppler perfusion index (DPI) increased significantly, indicating an improvement in hepatic blood flow as a consequence of the reduction in intrahepatic fat accumulation [123]. Serum AST remained stable, while ALT and γ-GT decreased [123]. These results agree with a second trial by Lewis et al. (2006); Spadaro et al. (2008); Oscarsson et al. (2018); and Green et al. (2020), in which treatment with n-3 PUFAs showed better serum biochemistry with reduction in TG, γ-GT and TNF-α levels, and improved insulin sensitivity [120,121,122,124]. In the same line, n-3 PUFAs as seal oil were administrated to patients with NAFLD associated with hyperlipidemia, resulting in lowered ALT and TG levels, normal liver echo patterns and half of them had a regression of fatty liver [158]. A study with the same objective was carried out for 3 months and showed that EPA and DHA plasma concentrations increase, as well as those of serum levels of adiponectin, with a reduction in TNF-α, LTB4, fibroblast growth factor 21 (FGF21), cytokeratin fragment 18 and PGE2 [132]. The authors concluded that n-3 PUFAs may benefit from the metabolic abnormalities associated with NAFLD exhibiting hyperlipidemia [132]. Olive oil enriched with n-3 PUFA was also studied for one year of supplementation, and patients with NAFLD had decreased circulating values of liver enzymes and TGs, with a significant improvement in the levels of adiponectin [159]. Additionally, purified EPA and DHA administration was investigated until an endpoint defined by a decrease in liver fat and the improvement of histologically validated biomarkers of liver fibrosis was attained. The results showed a decrease in the percentage of liver fat that can be achieved with a high percentage of DHA enrichment in erythrocytes from NAFLD patients [118,119], results that were confirmed by two subsequent studies [160,161]. NAFLD in pediatric patients was similarly considered in the context of supplementation with EPA plus DHA or DHA alone [133,134,135]. In all the studies, the authors reported an improvement in liver laboratory tests and a difference in fat/lean body mass after intervention, particularly DHA supplementation, which decreases liver and visceral fat and improves metabolic abnormalities in children with NAFLD [133], however, no changes in the fibrosis score were observed [136]. More recently, Parker et al. (2019) analyzed the effect of fish oil supplementation on liver and visceral fat in overweight men subjected to 12 weeks of supplementation with 1000 mg/day n-3 PUFAs [136]. This protocol did not alter liver fat, aminotransferases or visceral adiposity in overweight men, concluding that the dose used may not be sufficient to recommend for the sole purpose of reducing liver and/or visceral fat [136].

#### 7.2.2. NASH

From the studies that described NASH conditions in patients, in the first trial, highly purified EPA improves some NASH characteristics, including decreased serum ALT and AST levels as well as hepatic steatosis grade, with biopsy histology assessment showing improvement in several key features of NASH, such as hepatic steatosis, fibrosis, lobular inflammation and hepatocellular ballooning [162]. In particular, the authors found that the serum levels of pro-inflammatory cytokines such as TNFα tended to decrease after treatment, and the levels of the receptors TNF-R1 and sTNF-R2 also diminished [162]. Similar results were obtained by Nogueira et al. (2016), showing that increases in ALA and EPA are directly associated with better liver histology [137]. In the context of liver fibrosis, Li et al. (2015) examined whether n-3 PUFA administration for six months has an effect on NASH fibrosis parameters, where type IV collagen and pro-collagen type III were significantly diminished over control values [163]. In pediatric patients, DHA and vitamin E administration showed that the therapy is well-tolerated and can improve steatosis, ALT and fasting glucose levels [164]. It is important to mention, however, that Sanyal et al. (2014) [138] and Argo et al. (2015) [127] describe that EPA alone or combined EPA and DHA supplementation did not produce an improvement in the primary outcome of histological activity in patients with NASH.

#### 7.2.3. Hepatitis C

A clinical trial where EPA was administered to patients with chronic hepatitis C (HCV) resulted in an enhancement in EPA content of the erythrocyte membrane, lymphocyte counts increased after 4 weeks of therapy and maintained baseline throughout therapy and the serum ALT levels improved significantly [165]. The serum levels of 8-hydroxy-2’-deoxyguanosine at 24 weeks in the EPA group were significantly lower than in the controls, suggesting a reduction in DNA oxidation, which implies a beneficial effect of EPA supplementation in the treatment of patients with chronic hepatitis C [165]. Reinforcing the above, Takaki et al. (2008) showed that the administration of EPA diminished the hemolytic anemia induced by ribavirin (RBV) in the hepatitis C treatment. The control of this adverse effect is important to maintain the RBV treatment and avoid the sustained virological response, which is responsible for the most common viral cirrhosis and HCC [166]. Interestingly, administration of n-3 PUFA was associated with an enhanced efficacy of interferon-based therapies and greater antifibrogenic effect in obese patients with HCV [167]. 

#### 7.2.4. Liver Cirrhosis

Pazirandeh et al. (2007) conducted an assay where AA and DHA were supplemented in cirrhotic patients awaiting liver transplantation. The supplementation effectively raised the DHA levels in plasma lipid fractions and diminished the levels of n-6 PUFA derivatives, suggesting that DHA inhibited the elongation and desaturation of AA, which may have an anti-inflammatory outcome [168].

## 8. Supplementation with Specific Pro-Resolving Mediators (SPMs) in Liver Diseases

Exposure of the liver to different deleterious agents such as viruses, toxins and drugs determine alterations at the structural and functional level, generating chronic organ failure and liver cirrhosis in advanced stages. Nutritional therapy with essential fatty acids that are converted by the body into a novel genre of SPMs confer enhanced inflammation termination, tissue repair and regeneration [169]. Studies indicate that the production of these molecules is deregulated in several diseases, leading to a failure in resolution. It is noteworthy that numerous studies indicate that nutrition with n-3 PUFAs in the recommended daily doses generates an increase in several families of SPMs that correlate with improved responses of white blood cells in humans and reduce inflammation in mice [55].

### 8.1. SMP Effects in Experimental Animals

Among the SPM lipid autacoids, resolvins (Rv), are the best characterized, RvE1 being a drug candidate of the growing family of endogenous lipid mediators [169,170]. RvE1 reduces the levels of serum indicators of liver fibrosis, such as laminin, hyaluronic acid, pro-collagen type III and type IV collagen, slowing down the process of liver fibrosis, an effect that may be the result of adjustment of the immune and anti-inflammatory system [171]. Reinforcing the above contention, Kuang et al. (2016) demonstrated that RvD1 and RvE1 prevent concanavalin-A-induced liver damage and changes from hepatitis to liver cancer in mice by inhibiting transcription factors NF-κB and activating protein 1 (AP-1) as well as the respective inflammatory cytokine secretion [172]. In this context, it was recently shown that RvE1 and other eicosanoids were decreased in serum during NAFLD progression [173]. In relation to RvD1, an autacoid with anti-inflammatory and tissue restoration properties [174,175], when supplemented in a model of liver IR injury generates anti-inflammatory and resolutive effects in the liver. RvD1 attenuates IR-induced hepatocellular damage and the inflammatory response through promotion of anti-inflammatory M2 macrophages polarization, enhancement of AKT phosphorylation and/or regulation of mitochondrial redox homeostasis [176]. It is important to mention, however, that in a chronical model of cholestatic liver injury induced by bile duct ligation, the activity of hepatic stellate cells and the deposition of extracellular matrices were decreased by RvD1, without a notorious diminution in inflammation and the progression of fibrosis parameters [177]. Thus, RvD1 has limited therapeutic potential to treat cholestatic liver disease. Finally, the last SPM discovered, MaR1, also has important anti-inflammatory and resolution action on liver disease. In a murine model with carbon tetrachloride (CCl_4_)-induced liver damage, MaR1 attenuated liver damage, oxidative stress and lipid peroxidation [178]. MaR1 increased the activities of antioxidant mediators of the liver and decreased the production of inflammatory mediators such as TNFα, IL-6, IL-1β, monocyte chemotactic protein 1, myeloperoxidase, cyclooxygenase-2 and inducible nitric oxide synthase [178]. Furthermore, MaR1 inhibited the activation of NF-κB and MAPK in mouse liver, thus conferring antioxidant and anti-inflammatory properties in CCl_4_-induced liver injury [178]. In addition, MaR1 lowers lipogenic enzymes while inducing autophagy and fatty acid oxidation genes, which could be related to the activation of AMP-activated protein kinase (AMPK), an important regulator of cellular energy homeostasis [73]. More recently, Soto et al. (2020) showed in a murine model of liver IR injury that MaR1 generates activation of hepatocyte cell division via increased levels of IL-6 and nuclear localization of Nrf2, with decreased NF-κB activity [74]. These results open the possibility that MaR1 is a potential therapeutic tool in IR and other liver pathologies.

### 8.2. SPM Effects in the Clinical Setting

There is adequate evidence of the actions of the SPMs in animal models and several diseases or syndromes, including control of inflammatory pain [179], rheumatoid arthritis [180,181], diabetes [182,183,184] and atherogenesis [185], among others. However, studies related to the actions of SMPs on human liver disease have been scarcely explored. Previously, it was reported that supplementation with n-3 PUFAs to healthy voluntaries increased the concentration of plasma SPMs in relation to regulation of monocytes, neutrophiles and their inflammatory and adhesion molecules [186,187,188]. In relation to metabolic and liver disorders, Barden et al. (2015) described for the first time that subjects with metabolic syndrome had a diminution of resolvin precursors in plasma, and that the supplementation with n-3 PUFA increased total resolvin levels, except those of MaR-1 [189]. In this respect, the authors showed that when patients with metabolic syndrome lost weight (on average, body weight fell by 4.8 kg and reduction in waist circumference by 6.4 cm, described as a modest weight loss), neutrophil RvE1 clearly increased, but there were no significant changes in 18R-RvE3, RvD2 or Mar-1 [189]. Furthermore, Uno et al. (2016) showed that preoperative enteral immunonutrition reduced inflammatory responses and protected against the aggravation of postoperative complications in patients undergoing major hepatobiliary resection [154]. Furthermore, plasma levels of RvE1 were higher, whereas plasma levels of IL-6 decreased; therefore, RvE1 may play a key role in resolving acute inflammation when immunonutrition is supplemented by EPA [154]. More recently, Cata et al. (2017) demonstrated that the acute inflammatory response associated with liver resection in oncological patients (most of them where related to colorectal metastatic disease) is associated with an elevation in IL-6 and a depletion in RvD and lipoxin A4 at day one post-surgery, which tended to normalize at day five [190]. These studies allow understanding of the profiles of eicosanoids related to poor postoperative recovery and how to carry out pharmacological strategies to improve the resolution of acute inflammation related to liver disease, pointing to the need for more clinical trials to better understand the mechanisms of molecular action of these SPMs.

## 9. Conclusions

Research on nutrition with n-3 PUFAs has gained impetus with very encouraging results to improve the resolution of acute inflammation related to liver disease, due to their anti-inflammatory and cytoprotective properties. There is a large amount of information related to n-3 PUFA supplementation, with sufficient clinical trials to demonstrate their importance in liver disease. In addition, the supplementation with n-3 PUFAs as precursors of SPMs increase their tissue availability, therefore, studies have been carried out where it was possible to demonstrate a protective role in some liver diseases. However, evidence is still insufficient to be able to carry out a pharmacological strategy to improve the resolution of acute inflammation in liver diseases. This being the case, the situation calls for continuing clinical trials in order to better understand the mechanism of action of these lipid mediators.

## Figures and Tables

**Figure 1 nutrients-13-03384-f001:**
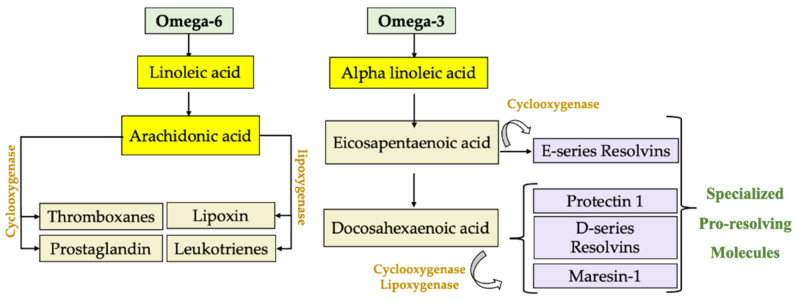
Endogenous production of lipid mediators using omega-6 fatty acids, where linoleic acid is transformed and arachidonic acid is generated, which is metabolized by lipoxygenase and cyclooxygenase to generate prostaglandins (PG), leukotrienes (LT), thromboxanes (TX) and lipoxins (LX). On the other hand, omega-3 alpha-linolenic acid is metabolized into eicosapentaenoic acid to generate resolvin E1 and docosahexaenoic acid to form protectin 1, resolvin-D and maresin-1, products with powerful cytoprotective and especially anti-inflammatory properties.

**Figure 2 nutrients-13-03384-f002:**
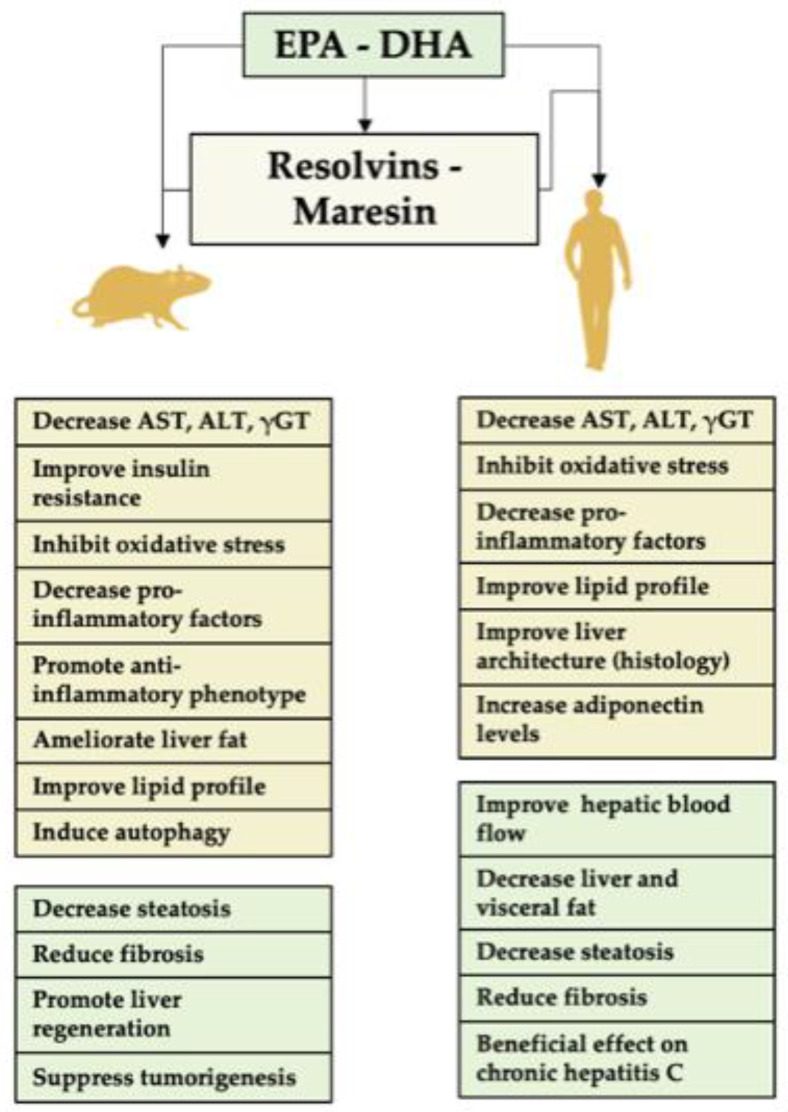
Role of EPA and/or DHA and their derivatives in liver disease. A summary of the activity of these fatty acids in animals’ assays and clinical trials is shown. The specific pro-resolving mediators (SPMs), resolvins and maresins, derived from EPA and DHA represent major agents for the prevention or resolution of liver inflammation. The underlying mechanisms include (i) diminution in liver polymorphonuclear leukocyte (PMN) infiltration by blocking PMN transendothelial migration; (ii) downregulation of transcription factor nuclear factor-κB (NF-κB) activity to decrease the expression of the pro-inflammatory cytokines TNF-α and IL-1β; and (iii) stimulation of macrophage (Kupffer cell) phagocytosis of apoptotic PMNs. EPA, eicosapentaenoic acid; DHA, docosahexaenoic acid; AST, aspartate aminotransferase; ALT, alanine aminotransferase; γGT, gamma glutamyl transferase; TNF-α, tumor necrosis factor-α; IL-1β, interleukin-1β.

**Table 1 nutrients-13-03384-t001:** Summary of official registers of clinical trials (in process or finished) related to the administration of omega-3 and liver conditions and/or disease *.

Liver Conditions or Diseases	Number of Studies	Most Common Dosage Used (Daily) **	Duration of Studios (Average in Months)	Others Clinical Conditions Evaluated or of Interest	General Analysis Realized or Proposed
NAFLD [117,118,119,120,121,122,123,124,125,126,127,128,129,130,131,132,133,134,135,136,137]	32	4 g (32%) ***	6 (0.5 to 24)	Three assays are related to morbid obesity and one of them to bariatric surgery One study is related to diabetics with PNPLA3 genotype	Liver parameters of injury measured by biochemical analysis and echography/tomography/MRI Lipids profile and liver fat fraction Insulin resistance Oxidative stress and inflammatory markers Abdominal adiposity (waist circumference)
NASH [125,138,139]	8	4 g (50%)	8 (0.75 to 12)	One study comprises long term parenteral immunonutrition related to hepatic steatosis	Liver parameters of injury measured by biochemical analysis and biopsy, echography/tomography/MRI Lipids profile and liver fat fraction Gut microbiota, abdominal adiposity (waist circumference) and other anthropometric measures
Hepatic disease related to parenteral nutrition [140,141,142,143,144,145,146,147,148,149,150,151]	22	1 g/kg/day (59%)	3 (1.5 to 60)	-	Liver parameters of injury measures by biochemical analysis General metabolic and fatty acid profile Inflammatory markers Gain of weight
HCC [88]	3	100 mL (10 g) (66.6%)	3	-	Postoperatory complications Hepatic and renal function Hospital stays Survival analysis
Healthy people [152]	2	0.5–1.2 g	1	-	Liver profile Changes in vitamin D and parathyroid hormone production Free fatty acids profile
Hepatitis C (chronic infection)	2	4 g	No specified	One assay related to insulin resistance	General biochemical analysis Analysis related to omega-3 as coadjuvant to interferon
Mayor liver resection [153,154,155]	2	100 mL (10 g)	1	One assay includes liver donor for transplant	Postoperatory complications Liver regeneration analysis
Hepatic metastasis [156,157]	4	4 g	12.5 (0.5 to 24)	Most of the studies are related to colorectal cancer	Quality of life General survival Changes in corporal lean fat Omega-3 distribution Ki67 and cancer proliferation Inflammation and angiogenesis
Liver transplants	3	1g/kg/day (75%) or 100 mL (10 g) (25%)	From the surgery until 5 days postoperatory and one month later	-	Analysis of neurologic complications Liver general parameters Allographic dysfunction Liver regeneration Survival analysis

*, The information was obtained from each of the Primary Registries authorized by the WHO and the Registry Network presented in https://www.who.int/ictrp/network/primary/en/and/ or presented in https://clinicaltrials.gov/ (on both webpages it was accessed from March to September 2021). **, Most of the studies do not detail the concentration of EPA and DHA separately. ***, The percentages in parentheses represent the universe of trials that used/are using those doses.

## Data Availability

Not Applicable.

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
