# Peer review of "N-3 Polyunsaturated Fatty Acids and Their Lipid Mediators as A Potential Immune–Nutritional Intervention: A Molecular and Clinical View in Hepatic Disease and Other Non-Communicable Illnesses"

_nutrients, 2021, doi:10.3390/nu13103384_

Round 1
Reviewer 1 Report
This is a very comprehensive and well written review of the topic. The title specifies hepatic disease but the review also addresses cardiovascular, nervous system, cancer, autoimmune diseases, etc. in section 6, thereby losing its focus. Either the title should be altered or the sections on these organ systems and diseases should be more limited or in an appendix (unfortunately since the information is excellent).
I have some minor suggestions for clarity and improvement.
Abstract: the last sentence comes off as somewhat ambivalent. Was this intended? (This sentence is also in past tense and does not coherent with the rest of the abstract.)
Line 75: “millions” should be “million” (This usage error occurs elsewhere as well e.g., line 89. Please check document elsewhere for similar issue.)
Lines 80-83: This is not a complete sentence but a series of phrases.
Line 84: “NALFD” should be “NAFLD”
Line 208: Please specify “in the dorsal area” of what organ or tissue? There is no context here.
Table 1: it would be helpful to provide the reference numbers in the table of the studies for each of the conditions listed. For example, not just list 8 studies for NASH but provide the numbers of the references from the Bibliography relating to that condition in case people wish to read the original work themselves.
Lines 428-431: Given that ribavirin is not used much now with medications such as sofosbuvir and other antivirals, is there any data on m-3 PUFA’s with those drugs in Hepatitis C?
Author Response
Dear Ms. Kang:
Thank you for reviewing our review article manuscript ID: 1361115 to be considered for publication in Nutrients. We have prepared a revised version of the work according to the comments of the two reviewers.
Reviewer-1:
1.- The title specifies hepatic disease but the review also addresses cardiovascular, nervous system, cancer, autoimmune diseases in section 6 thereby losing its focus.
Answer: We agree with this comment and decided to alter the title of the article to “N-3 polyunsaturated fatty acids and their lipid mediators as a potential immune-nutritional intervention: a molecular and clinical view in hepatic disease and other non-communicable illnesses”
Non-communicable illnesses” was also introduced in the Abstract, line 18.
2.- Abstract: the last sentence comes off as somewhat ambivalent. Was this intended (This sentence is also in past tense and does not coherent with the rest of the abstract).
Answer: This last sentence was eliminated and replaced by: “Since evidence, if still insufficient to carry out pharmacological trials to benefit the resolution of acute inflammation in non-communicable diseases, the foregoing calls for continuing preclinical and clinical research to better understand SPM actions and outcomes. “(lines 26-29).
3.- Lines 78-79: “millions” was replaced by “million”. This was also corrected in line 92.
4.- Lines 80-83: this is not a complete sentence but a series of phrases.
Answer: The sentence involved was eliminated.
5.- Line 86: “NALFD” was corrected to “NAFLD”.
6.- Line 208: Please specify “in the dorsal area” of what organ or tissue? There is no context here.
Answer: the line was rephrased for to better understanding.
7.- Table 1: it would be helpful to provide the reference numbers in the table of the studies for each of the conditions listed. For example, not just list 8 studies for NASH but provide the numbers of the references from the Bibliography relating to that condition in case people wish to read the original work themselves.
Answer: The references were added (only when the authors declared a clinical trial code, the rest was referenced in the text), Supplementary tables were added with more detail of clinical trial codes, references, and omega-3 composition.
8.- Lines 428-431: Given that ribavirin is not used much now with medications such as sofosbuvir and other antivirals, is there any data on n-3 PUFAʼs with those drugs in Hepatitis C?
Answer: Search in PubMed did not result in any publication concerning omega-3 and sofosbuvir. However, we found an article correlating omega-3 with interferon-based therapies, which was added to the text in lines 452-454:
“Interestingly, administration of n-3 PUFAs was associated with an enhanced efficacy of interferon-based therapies and greater antifibrogenic effect in obese patients with HCV [137].”
New reference 137 by Popa, S.G., Mota, M., 2007 was added to the reference list (lines 909-910), and the rest of the references were re-numbered in the text and reference list.
Reviewer-2:
1.- Despite its title, the discourse does not focus sufficiently on liver disease.
Answer: As indicated to reviewer-1, comment 1, the title of the work was changed to cover other pathologies given in section-6:
“N-3 polyunsaturated fatty acids and their lipid mediators as a potential immune-nutritional intervention: a molecular and clinical view in hepatic disease and other non-communicable illnesses.”
2.- My suggestion is to substitute paragraph 6 (Beneficial effects of n-3 PUFAs) with a paragraph describing the molecular mechanisms of omega-3 fatty acids on liver lipid metabolism.
Answer: We did not substitute paragraph 6, but added the requested new paragraph in section 7.1 (Evidence for n-3 PUFA effects in animal models of liver disease) in lines 339-346. New reference 102 was added to the list (lines 800-803) and the rest were re-numbered:
“The molecular mechanisms of n-3 PUFAs on liver lipid metabolism leading to an anti-steatotic response include (i) the activation of the PPAR-α/FGF21/AMPK/PGC-1α signaling cascade to increase FA oxidation; (ii) diminution in the processing of lipogenic factor sterol responsive element-binding protein 1c (SREBP-1c); (iii) downregulation of the expression of the citrate carrier reducing the cytosolic availability of acetyl-CoA units for de novo lipogenesis; and (iv) abrogation of endoplasmic reticulum stress and the associated expression of lipogenic enzymes [54,102].”
3.-In addition, the authors should consider adding a description of molecular mechanisms of omega-3 on liver inflammation.
Answer: As requested, the following new paragraph was introduced in lines 188-196:
“The molecular mechanisms involved in the anti-inflammatory effect of n-3 PUFAs in the liver are related to the downregulation of pro-inflammatory transcription factor nuclear factor-κB (NF-κB). These include (i) formation of SPMs; (ii) binding to peroxisome proliferator-activated receptor-α (PPAR-α) to form an inactive complex with NF-κB-p60; (iii) binding to G-protein coupled receptor 120 (GPR120) to abrogate NF-κB activation and derived inflammasome NLRP3 expression; (iv) formation of J3-isoprostanes, oxidation products that activate nuclear factor erythroid-2-related factor 2 (Nrf2) that induces an antioxidant response blocking the redox activation of NF-κB [54].”
New reference 54 was added to the list (lines 676-678) and the rest of them were re-numbered.
4.- Before paragraph 8, the authors should add a paragraph discussing the molecular mechanism of SPMs on liver disease.
Answer: We consider adding the requested paragraph in the legend of Figure 2, which refers to the actions of SPMs in the liver:
“The specific pro-resolving mediators (SPMs) resolvins and maresin derived from DHA represent major agents for the prevention or resolution of liver inflammation. The underlying mechanisms include (i) diminution in liver poly-morphonuclear leukocyte (PMN) infiltration by blocking PMN transendothelial migration; (ii) downregulation of transcription factor nuclear factor-κB (NF-κB) activity to decrease the expression of the pro-inflammatory cytokines TNF-α and IL-1β; (iii) decrease in TNF-α-dependent PMN superoxide radical (O2.-) generation; and (iv) stimulation of macrophage (Kupffer cell) phagocytosis of apoptotic PMNs.” (Lines 328-333).
5.- Minor comments:
Table 1 please add the composition of omega 3 fatty acids used in clinical trials:
Answer: Supplementary tables were added to better detail n-3 PUFA composition.
All changes introduced are indicated in yellow.
Best regards
Jessica Zúñiga-Hernández, PhD.
Departamento de Ciencias Básicas Biomédicas
Facultad de Ciencias de la Salud
Universidad de Talca,
Talca, Chile
email: [email protected]
Reviewer 2 Report
I have reviewed the article tilted: “N-3 polyunsaturated fatty acids and their lipid mediators as potential immune-nutritional intervention in hepatic disease: a molecular and clinical view.”
This review is well written, and the authors discuss an interesting topic; however, despite its title, the discourse does not focus sufficiently on liver disease.
Major comments:
The authors discuss the beneficial effect of two important molecules: omega-3 fatty acids and specialized po-resolving mediators, on liver diseases.
- My suggestion is to substitute paragraph 6 (Beneficial effects of n-3 PUFAs) with a paragraph describing the molecular mechanisms of omega-3 fatty acids on liver lipid metabolism.
- In addition, the authors should consider adding a description of molecular mechanism of omega-3 fatty acids on liver inflammation.
- Before paragraph 8 (Supplementation with specific pro-resolving mediators (SPMs) in liver diseases), the authors should add a paragraph discussing the molecular mechanism of SPMs on liver disease.
Minor comments
Table 1 please add the composition of omega3 fatty acids used in clinical trials
Best regards
Author Response

(The authors gave the same response as above.)

Round 2
Reviewer 2 Report
No further comments.